# Comprehensive Sequence Analysis of Parvalbumins in Fish and Their Comparison with Parvalbumins in Tetrapod Species

**DOI:** 10.3390/biology11121713

**Published:** 2022-11-25

**Authors:** Johannes M. Dijkstra, Yasuto Kondo

**Affiliations:** 1Center for Medical Science, Fujita Health University, Dengaku-gakubo 1-98, Toyoake 470-1192, Japan; 2Department of Pediatrics, Fujita Health University Bantane Hospital, Otobashi 3-6-10, Nakagawa, Nagoya 454-8509, Japan

**Keywords:** parvalbumin, oncomodulin, fish, evolution, sequence, structure

## Abstract

**Simple Summary:**

If people are allergic to fish meat, this usually is because of an immune reaction against parvalbumin proteins in fish muscle. Fish have multiple parvalbumins with quite similar sequences, making a comparison between fish species complex. The present study is the first to do so in a thorough manner so that researchers interested in fish allergies can more readily understand the underlying molecular biology. In addition, we look at ancient sequence motifs that distinguish the major types of parvalbumins—such as oncomodulins and α-parvalbumins—that are found in both fish and tetrapod species. This should help researchers interested in parvalbumin functions to better understand the essence of each type of parvalbumin.

**Abstract:**

Parvalbumins are small molecules with important functions in Ca^2+^ signaling, but their sequence comparisons to date, especially in fish, have been relatively poor. We here, characterize sequence motifs that distinguish parvalbumin subfamilies across vertebrate species, as well as those that distinguish individual parvalbumins (orthologues) in fish, and map them to known parvalbumin structures. As already observed by others, all classes of jawed vertebrates possess parvalbumins of both the α-parvalbumin and oncomodulin subfamilies. However, we could not find convincing phylogenetic support for the common habit of classifying all non-α-parvalbumins together as “β-parvalbumins.” In teleost (modern bony) fish, we here distinguish parvalbumins 1-to-10, of which the gene copy number can differ between species. The genes for α-parvalbumins (*pvalb6* and *pvalb7*) and oncomodulins (*pvalb8* and *pvalb9*) are well conserved between teleost species, but considerable variation is observed in their copy numbers of the non-α/non-oncomodulin genes *pvalb1-to-5* and *pvalb10*. Teleost parvalbumins 1-to-4 are hardly distinguishable from each other and are highly expressed in muscle, and described allergens belong to this subfamily. However, in some fish species α-parvalbumin expression is also high in muscle. Pvalb5 and pvalb10 molecules form distinct lineages, the latter even predating the origin of teleosts, but have been lost in some teleost species. The present study aspires to be a frame of reference for future studies trying to compare different parvalbumins.

## 1. Introduction

A considerable number of people are allergic to fish meat, in which parvalbumins are the major allergens that, in extreme cases, can even provoke death [1]. However, how the different parvalbumins in a single fish species are related to each other, to parvalbumins in other fish species, and to parvalbumins in tetrapod species, is poorly understood.

Parvalbumins are small molecules, mostly comprising 108–109 amino acids, that can bind calcium ions with high affinity. There is a strong Ca^2+^ concentration gradient across the membranes of living cells, and cells need to protect themselves by keeping most Ca^2+^ outside. However, cells can bear some fluctuation in internal Ca^2+^ concentrations, and cross-membrane Ca^2+^ gradients can be used in signaling pathways such as in neurons and muscle cells [2]. Most, but not all, functions described for parvalbumins are related to controlling/modifying this type of signaling, in which parvalbumins are generally believed to not functionally interact with other proteins but merely act as a slow Ca^2+^ buffer [3]. Parvalbumins belong to a large superfamily of “EF-hand” domain-containing proteins in which two α-helices are bridged by a Ca^2+^-chelation loop; parvalbumins have two such domains, comprising their helices C plus D and their helices E plus F (for a structure see Figure 1A). Among the other members of this superfamily are, for example, calmodulin, troponin C, and calretinin [4,5,6,7]. When the Ca^2+^ concentration is low, parvalbumins can bind Mg^2+^ ions for which they have a lower affinity, and different exchange rates of Mg^2+^ for Ca^2+^ are important for the functional dynamics of each specific parvalbumin [8,9].

Parvalbumins were first found in tissues where they are abundant, namely in the muscles of ectotherm vertebrate species like frogs [12] and fish [13,14]. Already early on, it was clear that single species can have multiple different forms of these molecules, and the family name “parvalbumins” was chosen because of their small size (*parvus* is Latin for small) and their solubility in water akin to albumin [14]. However, parvalbumins are not related to albumins and, in contrast to serum albumins, parvalbumins are typically (but not always) cytosolic. The first parvalbumin sequences were determined for the fishes carp [15,16], hake [17], and pike [18,19], and the first structure of a parvalbumin molecule was determined for a parvalbumin isolated from carp muscle [11]. Later, parvalbumins were also found in reptiles, birds, and mammals including humans [20,21].

Traditionally, two lineages of parvalbumins have been distinguished, alpha and beta, with the beta parvalbumins being the more acidic ones [22,23]. The two “lineages” share around 50% amino acid identity and already existed in early jawed vertebrates [4,24,25,26]. We agree that there is a clear alpha lineage, but the present study disputes that there is convincing evidence for all the other sequences being clumped together as a phylogenetic sister lineage of this alpha lineage. Therefore, we try to avoid the term “beta parvalbumins.”

Humans have one gene for an α-parvalbumin—often simply called “parvalbumin” but in the present study called “α-parvalbumin” to avoid confusion—and two genes for non-α parvalbumins, the nearly identical “oncomodulin-1” and “oncomodulin-2.” Mammalian α-parvalbumin is a cytosolic protein and is used as a marker for several neuron subsets [27,28,29], in which it affects the rate of neural signaling by being a slow buffer of Ca^2+^ [30,31]. Furthermore, mammalian α-parvalbumin has been found in fast-twitch muscle fibers, parathyroid glands, and distal convolute tubules in nephrons [32,33,34,35]. Notably, however, in human muscles, in contrast to rodents, α-parvalbumin amounts are very low and close to undetectable ([36]; also see The Human Protein Atlas, available online: https://www.proteinatlas.org/ (accessed on 27 October 2022)). In muscles, the influx of Ca^2+^ from the sarcoplasmic reticulum into the sarcoplasm induces contraction, and in fast-twitch muscle fibers of rodents, the binding of Ca^2+^ by cytosolic α-parvalbumin after contraction increases the muscle relaxation rate—hence shortening the time of a contraction-relaxation cycle—so that α-parvalbumin is believed to support rapid, phasic movements [8,36,37,38]. Mammalian oncomodulin is found in certain tumors—which together with its resemblance to calmodulin inspired its name [39,40]—as well as in the cytotrophoblast of the fetal placenta [41], sensory hair cells in the inner ear [42], and some immune cells [43]. Cytosolic oncomodulin in hair cells in the inner ear contributes to proper ear functioning [44], and oncomodulin secreted by macrophages has a regenerative effect on neurons [43,45]; however, the mechanisms of these functions have not been well clarified yet [46].

For chickens, three different parvalbumin molecules have been described at the protein level and were named chicken parvalbumin 1 and -3 (CPV1 and -3) and avian thymic hormone (ATH) [47,48,49,50]. CPV3 and ATH expressions are high in the thymus, where they are expressed by stromal cells, and they are believed to be secreted molecules that stimulate immune cells [47,50,51,52]. CPV3 is homologous to the oncomodulins in mammals, and—reminiscent of those—is found in hair cells of the avian auditory organ (basilar papilla) [53]. Chicken ATH, besides in the thymus, is also expressed in the eye, and, hitherto, was believed to not have an equivalent in mammals while being similar to non-α-parvalbumin in amphibians and reptiles [54,55]. CPV1 is the single α-parvalbumin in chicken [25] and is present in muscle [48,54]. Like in chickens, in sharks the parvalbumins abundant in muscle tissue are α-parvalbumins [56,57]. In frogs, both α-parvalbumins and non-α/non-oncomodulin-parvalbumins are abundant in muscle tissue [58,59,60,61,62] and are present together in the same muscle cells [63].

Teleost fish (modern ray-finned bony fish) have multiple types of parvalbumins with different expression patterns ([26,64] and see ZFIN, available online at https://zfin.org/ (accessed on 27 October 2022)), but most studies have been on the parvalbumins that are abundant in fish muscles and can be allergenic by provoking IgE-type antibody responses in sensitive people (e.g., [65]). These fish muscle parvalbumins have been classified into the beta lineage together with oncomodulins and other non-α-parvalbumins in tetrapod species [22,23], but the present article disputes the reliability of this classification. It is not well known why these fish molecules are such strong allergens, but at least partially it appears to be explained by their high abundance and unusual stability [1]. Generally, patients allergic to teleost fish parvalbumin show high cross-reactivities against the meat of several teleost fish species but not of others [1,66,67,68], while some studies even reported their cross-reactivities against chicken α-parvalbumin [69], crocodile β-parvalbumin [70], frog α-parvalbumin and β-parvalbumin [71], and shark and ray α-parvalbumin [72].

Currently, a thorough comparison between the sequences of different parvalbumins is lacking, which hampers the quality of discussions regarding their functional or allergenic properties. Therefore, the last author of this study, YK, who specializes in allergic reactions (e.g., [67,73]), asked the first author, JMD, who is a specialist in fish gene evolution (e.g., [74,75]), to provide a comprehensive sequence analysis. The present study aims to help future studies to more easily understand the differences between different parvalbumins.

## 2. Materials and Methods

### 2.1. Retrieving of Genes

Parvalbumin genes were searched in NCBI (GenBank; Available online https://www.ncbi.nlm.nih.gov/ (accessed on and before 27 October 2022)) and Ensembl (Available online http://asia.ensembl.org/index.html (accessed on and before 27 October 2022)) databases based on their annotations and, also, by BLAST homology searches. In some cases, the database exon predictions from genomic sequences were adjusted to agree better with parvalbumin consensus sequences (see Appendix A). For the identification of neighboring genes, in most cases, the database annotations were followed, while in some cases we performed similarity searches to identify genes.

### 2.2. Estimation of Transcription Levels

For estimation of transcription levels of fish parvalbumin genes, the relative numbers of sequence matches were compared between single read archive (SRA) datasets for different tissues. See Appendix A for a detailed explanation.

### 2.3. Sequence Alignment

Parvalbumin sequences were aligned by hand (see Appendix A), based on similarities and the estimated likelihood of evolutionary events (as in [76]). Consensus sequence patterns were determined by WebLogo software (Available online: https://weblogo.berkeley.edu/logo.cgi (accessed on 1 October 2022)).

### 2.4. Phylogenetic Tree Analysis

Phylogenetic tree analysis was performed with MEGA 7.0 software (Available online: https://mega.software.informer.com/7.0/ (accessed on and before 27 October 2022)). For details see Appendix A.

### 2.5. Analysis and Depiction of Molecular Structures

For analysis and depiction of molecular structures from the PDB database, PyMOL 2.5.2 software (Available online: https://pymol.org/2/ (accessed on and before 27 October 2022)) was used.

## 3. Results and Discussion

### 3.1. The Organization of Our Study to Compare Parvalbumin Sequences with a Focus on Teleost Fish

Teleost fish form the largest group of fish by far and include most of the commercial fish species [77]. Shortly after their evolutionary separation from Holostei (which include gars and bowfin), their ancestor experienced a whole-genome duplication (WGD) event around 330 million years ago (MYA) [78]. An important study by Mukherjee et al., 2021 [26], which we used as inspiration, although those authors only analyzed nucleotide sequences and we do not agree with some of their conclusions, showed how this WGD resulted in the duplication from two to four genomic regions with parvalbumin genes (see also Figure 2). We selected 16 representative teleost species for investigating all their parvalbumin sequences: European eel (*Anguilla anguilla*), elephant fish (*Paramormyrops kingsleyae*), zebrafish (*Danio rerio*), *Onychostoma macrolepis* (this species has no English name), Atlantic herring (*Clupea harengus*), channel catfish (*Ictalurus punctatus*), Northern pike (*Esox lucius*), and the neoteleosts Atlantic cod (*Gadus morhua*), fugu (*Takifugu rubripes*), yellowtail amberjack (*Seriola lalandi*), greater amberjack (*Seriola dumerili*), turbot (*Scophthalmus maximus*), gilthead seabream (*Sparus aurata*), barramundi perch (*Lates calcarifer*), Nile tilapia (*Oreochromis niloticus*), and Japanese medaka (*Oryzias latipes*) (Appendix A). For ten of these 16 species (we considered three species to be sufficiently representative for the nine investigated neoteleosts) the synteny of parvalbumin-containing genomic regions is shown in Figure 2, and approximations of their tissue-specific parvalbumin RNA expression levels—according to database analysis—are summarized in Appendix A, except for yellowtail amberjack for which we deemed the available datasets not sufficiently complete. For comparison, sequence analysis of all parvalbumins was also performed for the primitive fishes spotted gar (*Lepisosteus oculatus*), gray bichir (*Polypterus senegalus*), and elephant shark (*Callorhinchus milii*), and for the lobe-finned/tetrapod species coelacanth (*Latimeria chalumnae*), tropical clawed frog (*Xenopus tropicalis*), chicken (*Gallus gallus*), platypus (*Ornithorhynchus anatinus*), Tasmanian devil (*Sarcophilus harrisii*), Iberian mole (*Talpa occidentalis*), mouse (*Mus musculus*), and human (*Homo sapiens*) (Appendix A; representative species in Figure 2). Additionally, for various reasons, we also included some individual parvalbumin sequences from a variety of species in the sequence comparison (Appendix A). The deduced amino acid sequences of 210 parvalbumins are aligned in Appendix A, and the sequence logos—which highlight shared sequence characteristics—for all 210 parvalbumins and per category are shown in Figure 3. For the sequences aligned in Appendix A, phylogenetic trees were created by several methods (Figure 4 and Appendix A).

In the ancestors of some commercially important fishes such as salmonids (e.g., Atlantic salmon [*Salmo salar*]) and common carp (*Cyprinus carpio*) there has been an additional, relatively recent, WGD event. This has further increased the number of parvalbumin genes in these species (with 22 found in common carp) and complicated their analysis, as discussed by Mukherjee et al. [26]. The present study does not include the complete analysis of the parvalbumin genes of these species because there are sequence reliability issues and differences between multiple whole-genome sequence reports for the same species, and the extra complexity would interfere with the clarity of our messages.

In jawless fish (Agnatha) and invertebrates, there are no sequences that can univocally be recognized as “parvalbumins” ([26] and our unpublished data) and the present study does not investigate those species. As outgroups for our phylogenetic tree analyses, we only included one calmodulin sequence and two teleost fish representatives of a family of “parvalbumin-like” molecules that is widely distributed throughout jawed vertebrates (e.g., see GenBank accessions NP_001341568 and XP_041072779 for human and great white shark parvalbumin-like sequences, respectively). Future studies may try to assess how and when in evolution the different families of EF-hand domain-containing proteins emerged. The present study does not aim to provide an extensive overview of parvalbumin sequences in tetrapod species, but nevertheless, in various aspects, probably has more information on this than previous studies. Our study also did not delve deeply into the parvalbumin situation in Chondrichthyes (sharks/rays and chimeras).

**Figure 3 biology-11-01713-f003:**
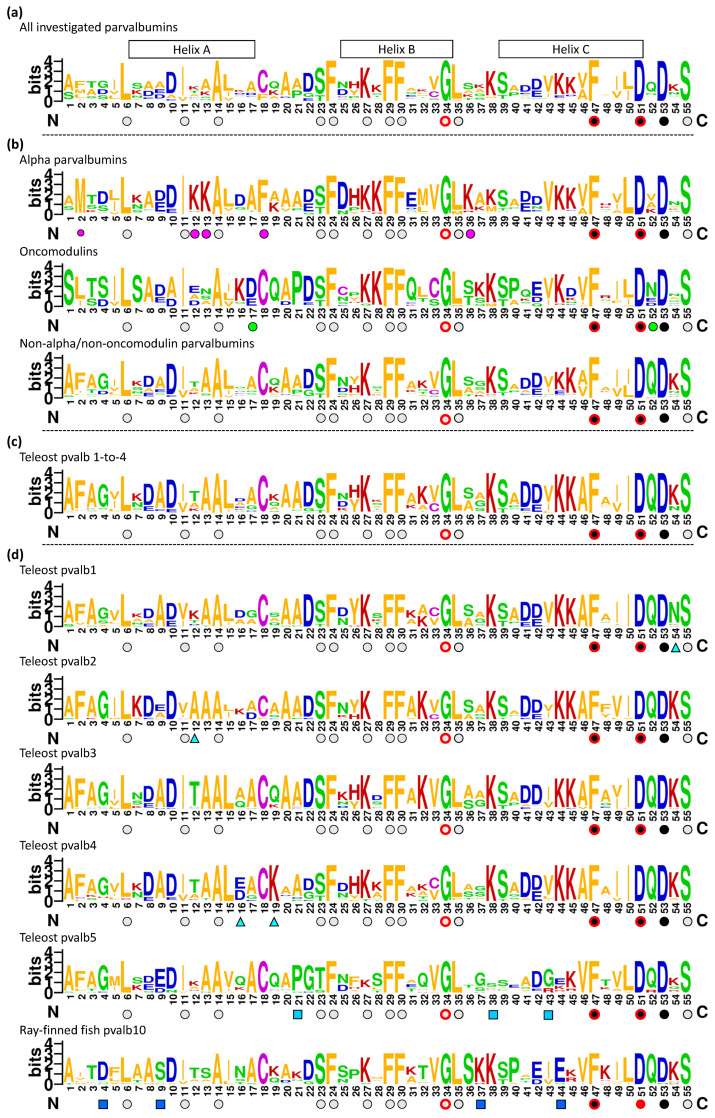
Parvalbumin consensus sequences. This is a two-page figure with residues 1–55 shown on the first page and residues 56–109 on the second. Sequence logos are shown for the parvalbumin sequences as compared in the Appendix A alignment figure (**a**) for all sequences, (**b**) per α-parvalbumin, oncomodulin, and non-α/non-oncomodulin categories, (**c**) for the combined teleost pvalb1-to-4 sequences, and (**d**) for parvalbumins encoded by individual teleost genes pvalb1, -2, 3, 4, 5, and -10. The sizes of the single letters that represent amino acids correspond with their level of conservation. The symbols below the sequences highlight, somewhat arbitrarily, notable high levels of conservation of residues or similar residues: Black circles, EF-hand domain molecules (following [80]); gray circles, (more specifically) parvalbumins; magenta circles, α-parvalbumins; green circles, oncomodulins; blue circles, non-α/non-oncomodulin parvalbumins; cyan triangles, rather characteristic of teleost pvalb1, -2, or -4; light-blue squares, teleost pvalb5; dark-blue squares, teleost pvalb10. Red lines around black or gray circles indicate that the residue is conserved in all the 210 investigated parvalbumin sequences (Appendix A).

**Figure 4 biology-11-01713-f004:**
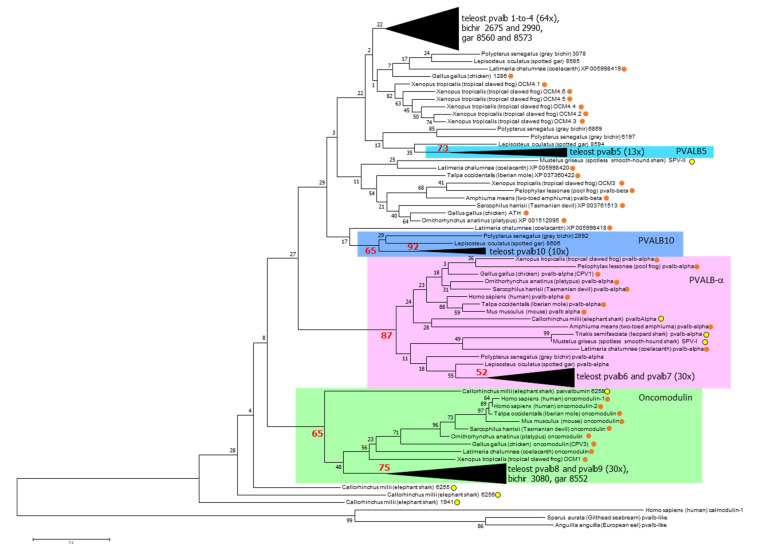
Molecular Phylogenetic analysis by Maximum Likelihood method. The percentage of trees in which the associated taxa clustered together is shown next to the branches, and important values >50 are highlighted in large red font. The tree is drawn to scale, with branch lengths measured in the number of substitutions per site. Clusters with teleost sequences (including sometimes also non-teleost sequences) are condensed; for a non-condensed tree and full descriptions see Appendix A. This type of analysis can not clearly distinguish between teleost pvalb1, -2, -3, and -4, between teleost pvalb6 and -7; and between teleost pvalb8 and -9 (Appendix A; see also [26]). Yellow and orange circles highlight parvalbumin sequences of sharks/rays/chimeras and lobe-finned-fish/tetrapods, respectively.

### 3.2. The Overall Phylogeny of Parvalbumin Sequences in Jawed Vertebrate Species; There Are Ancient Alpha and Oncomodulin Lineages, but Their Relationships to the Other Parvalbumins Cannot Be Assessed with Certainty

Phylogenetic tree analyses readily distinguish ancient α-parvalbumin and oncomodulin lineages that are represented in all clades of investigated jawed vertebrates (Figure 4 and Appendix A). This agrees with findings by, for example, Modrell et al., 2016 [25], and Mukherjee et al., 2021 [26], who both named the oncomodulin lineage the “β1” lineage. The α-parvalbumin sequences have a relatively large number of unique features that are well conserved among them (see Section 3.6.1). However, this does not compellingly identify the α-parvalbumins as an evolutionary sister-lineage to the combination of all other parvalbumins—in such model classified as the “β lineage”—as initially proposed based on biochemical properties and early types of phylogenetic tree analysis using only a few parvalbumin sequences [22,23]. None of the phylogenetic tree methods that we applied supports the α-β sister lineage concept (Figure 4 and Appendix A), and Friedberg already declared in 2005 [64] that this sister-lineage idea is “not sustainable.” However, most studies have ignored this and maintained the α-β concept and/or nomenclature, even though tree analysis by Modrell et al., 2016 [25], suggests (although with low bootstrap values) that the oncomodulins form a sister lineage to the combined α-parvalbumins plus what they call “β2” parvalbumins (all parvalbumins which are not α or β1). Recently, Mukherjee et al., 2021 [26], did find phylogenetic tree support for α, “β1” and “β2” lineages by analyzing nucleotide sequences, but when estimating deep phylogenies their type of analysis may lack sensitivity because of encountering nucleotide substitution “saturation” issues [81] while the Bayesian method applied may have been “overconfident” in the estimation of branch reliabilities [82,83]; our unpublished nucleotide-sequence based tree analyses using different methods could not confirm their results for deep parvalbumin branches, and some of their supposedly well-supported younger branches do not agree with gene locations. For example, Mukherjee et al. [26] reported the two European eel oncomodulin sequences *pvalb8* and *pvalb9* (see our Figure 2) to phylogenetically cluster together apart from *pvalb9* orthologues in other teleost fishes with a confidence of 0.84.

Looking at the combined data of our present study and [25,26], a reasonable, though not certain, estimation seems to be that at least the non-α/non-oncomodulin genes situated in the human-Chr.7-equivalent regions of ray-finned fish and lobe-finned-fish/tetrapods (Figure 2) are members of a single lineage that is separate from the α-parvalbumin and oncomodulin lineages (with unknown branching order between those three lineages). However, this lineage may not include non-α/non-oncomodulin sequences of primitive elephant sharks (Figure 4). Basically, the fact that parvalbumins have short and well-conserved sequences interferes with the resolution power of phylogenetic tree analysis, and the deepest parvalbumin lineages and their branching cannot be determined with certainty.

### 3.3. The Parvalbumin Genes Map to Paralogous Regions That Derived from Whole-Genome Duplication Events

The human chromosomes harboring oncomodulin and α-parvalbumin genes are Chr. 7 and Chr. 22, respectively (Figure 2), and these and their corresponding regions in other jawed vertebrates represent paralogous regions that derived from a WGD event early in the vertebrate lineage [84]. Because of an extra WGD event, these regions have extra duplicates in teleost fishes (Figure 2). In the species that we investigated, at least one oncomodulin gene (green in Figure 2) and one α-parvalbumin gene (magenta in Figure 2) were stably maintained in the human-Chr.7-equivalent and human-Chr.22-equivalent regions, respectively. In Figure 2, the non-α/non-oncomodulin parvalbumin genes are colored blue, and they differ between zero and seven in the investigated human-Chr.7-equivalent regions and between zero and two in the human-Chr.22-equivalent regions (Figure 2). Most eutherian (placental) mammals, including human and mouse, appear to have lost genes for non-α/non-oncomodulin parvalbumins, but such genes can be found in the human-Chr.7-equivalent regions in Eulipotyphla (a clade of eutherian mammals including shrews, moles, and hedgehogs; represented by Iberian mole in Figure 2) and in monotremes (like platypus) and marsupials (like the Tasmanian devil) (Figure 4 and Appendix A). To the best of our knowledge, the presence of non-α/non-oncomodulin parvalbumins in mammals has not been described previously. Mukherjee et al., 2021 [26], proposed that “α and β” lineages only were established in the human Chr. 22- and Chr. 7-equivalent regions, respectively, after the WGD event in an early vertebrate, followed by the development of “β1” (oncomodulin) versus “β2” (rest) lineages resulting from a tandem gene duplication in the human-Chr.7-equivalent region. However, that model is inconsistent with their own assignment as “β2” lineage members of those non-α/non-oncomodulin parvalbumin genes that were found in the human-Chr.22-equivalent regions in elephant shark scaffold KI636125 (“1941” in Figure 2) and bichir Chr. 10 (“6869” and “6197” in Figure 2).

### 3.4. Parvalbumin Nomenclature and Lineages in Teleost Fish; Delving beyond the Concept of “Microheterogeneity”

Because our study focuses on teleost fish, only for these species do we here propose a partially new (and hopefully final) parvalbumin nomenclature. This nomenclature is mostly based on gene orthologies with the “parvalbumin 1-to-9” (pvalb1-to-9) genes identified in zebrafish by Friedberg, 2005 [64], and follows the numerical designations in that study. This necessitates the creation of the new name “parvalbumin 10” for an ancient gene that was lost in zebrafish but conserved in many other teleosts. In cases where gene duplications occurred after the ancestors of the respective species and zebrafish separated in evolution, we add a letter to the name starting from A (e.g., parvalbumins 2A and 2B in Northern pike); the same added letters in different species do not necessarily relate to the same duplication event (e.g., parvalbumins 2A and 2B in Northern pike and Atlantic herring derived from independent pvalb2 gene duplications). In teleost species whose ancestors underwent an extra WGD event, such as common carp and Atlantic salmon [85], we added a chromosome indication between brackets to the parvalbumin name [e.g., Atlantic salmon pvalb4_(Chr.3)].

In teleost fish, five parvalbumin lineages can be readily distinguished by phylogenetic tree analysis (Figure 4 and Appendix A): (1) the α-parvalbumins (pvalb6 and pvalb7); (2) the oncomodulins (pvalb8 and pvalb9); (3) the muscle parvalbumins pvalb1-to-4; (4) pvalb5; and (5) pvalb10. These lineages were also observed in the phylogenetic tree analysis by Mukherjee et al., 2021 [26], who, like us, also found that phylogenetic tree analysis cannot properly distinguish between pvalb6 and pvalb7, between pvalb8 and pvalb9, and between pvalb1, -2, -3, and -4. This inability is probably caused by these molecules being generally well-conserved and having relatively few variable residues whose identities seem to somewhat “fluctuate around a conserved theme” rather than develop in a certain direction. In the present study, we could nevertheless identify the individual teleost parvalbumin genes by additionally looking at genomic locations, by assessing the detailed gene evolution through the study of closely related species, and by finding distinguishing sequence motifs. Gene identification was easy for the genes *pvalb6-to-9* because they are single genes at conserved genomic locations (Figure 2). For *pvalb5* and *pvalb10,* it was also not so difficult, although absent in some fishes, because of their distinctive characteristics (Figure 3 and Figure 4, Appendix A). However, it was quite a puzzle for the genes *pvalb1-to-pvalb4*—even though *pvalb1* + *pvalb4* can readily be distinguished from *pvalb2* + *pvalb3* because of their genomic locations (Figure 2)—and some of those assignments should be considered educated guesses, based on considerations of parsimony regarding, for example, gene locations and characteristic residues, rather than evolutionary certainties. The difficulty explains why the identities of muscle parvalbumins across fish species have typically been ignored and their variation has been summarized by others as “microheterogeneity” [86] or “high isoform complexity” [87], although a deeper level of understanding is possible.

### 3.5. The Consensus Sequence of Parvalbumins Mapped to the Molecular Structures

The sequence logos in Figure 3 show that a staggering 42 residues are very well (though not perfectly) conserved between parvalbumins. In this chapter, we analyze their locations in the parvalbumin molecular structure. The pronounced structural features of parvalbumins, including the Ca^2+^ binding sites and the hydrophobic core, were already recognized from the first structural analyses [5,11], but in order to be comprehensive, we show them here again and present them in the context of residue conservation. In 2004, Bottoms et al. [88] stated that “*Despite our familiarity with the parvalbumin molecule, however, our understanding of parvalbumin structure–function relationships remains embryonic*”. Since then, the situation has not much changed, and it is not really known how parvalbumin structures and functions are related. The binding of Ca^2+^ stabilizes the parvalbumin structures, but even if the CD and EF binding sites are free of metal ions, the overall parvalbumin structures (unless denaturated) are sometimes not very different, and α-parvalbumins especially are quite stable (summarized in [89]). It has been speculated that the hydrophobic core of parvalbumins somehow participates in transferring information between the two cation binding sites [88,90]. However, the structural comparisons in the present study are primarily intended to provide a topographical structural understanding of the sequences, and we will mostly stay away from speculations on function as they are not our field of expertise and most speculations would be rather vague anyway. Nevertheless, we do point out that some structural “plasticities” that have been interpreted by others as indicative of functional mechanisms may be primarily attributable to whether the structure was determined by X-ray crystallography or solution nuclear magnetic resonance (NMR) and could, at least to some extent, represent methodological artifacts instead (see Section 3.5.4).

#### 3.5.1. Similarities between and within Parvalbumins

Figure 1B shows superimpositions of the structures of a variety of representative parvalbumins including three α-parvalbumins, two oncomodulins, and five other parvalbumins. This reveals a common overall structure (e.g., [5,91]), which is consistent with the investigated parvalbumins having a large consensus motif of well-conserved residues (Figure 3A). Figure 3 and Appendix A show the mature proteins without the starting methionine which typically is cleaved off and without an indication of the commonly found N-terminal acetylation (e.g., [21]). In Figure 5A, as in Figure 3, residues that are well conserved between parvalbumins and several other molecules with paired EF-hand domains [80] are highlighted with black, and other residues that are very well conserved across parvalbumins are highlighted with gray; in Figure 5A, showing carp pvalb4_(Chr.A3), the sidechains of these residues are shown in sticks format, revealing that most of the “black” residues are associated with the Ca^2+^ binding structures. Some residues are perfectly conserved in all the 210 parvalbumins that we compared (Appendix A), which in Figure 3 is highlighted by red-lined circles. The structures with helices C + D and helices E + F are clearly derived from genetic duplications, and it has been speculated that the parvalbumin region with the A + B helices is also related to the other helical regions [92]. However, we have not seen sufficient evidence for assuming the latter (see also [93,94]). Figure 5B shows a superimposition of the CD and EF helices containing structures of common carp pvalb4_(Chr.A3), highlighting the same conserved residues as in Figure 5A but using different colors for the CD and EF regions and their residues. Figure 5B shows how three acidic residues involved in binding Ca^2+^ are conserved between the CD and EF regions, namely D51/D90, D53/D92, and E62/E101. The other two residues that are well conserved between the CD and EF regions are G56/G95 and I58/I97, which are also part of the Ca^2+^ binding structure (Figure 5B). The other residues that are conserved among parvalbumins show less similarity between the CD and EF regions, although in two cases at similar positions different well-conserved hydrophobic residues can be found, namely L63 and L67 in helix D and F102 and V106 in helix F (Figure 5B).

#### 3.5.2. The Binding of Ca^2+^

Figure 5B also shows that parvalbumin CD regions use E59 to help bind Ca^2+^, whereas the EF regions do not have a similar residue at this position but commonly have a glycine (G98, see Figure 3); this type of difference is consistent with the fact that the two Ca^2+^ binding sites can have different ion affinities [5]. Their structures are better shown in Figure 5C,D, which include depictions of the polar contacts formed by Ca^2+^ when bound to the CD and EF binding sites; the depicted structures are of carp pvalb4_(Chr.A3) but their principle is common among investigated parvalbumins (e.g., [95,96]).

#### 3.5.3. The Conserved Inner Core

Figure 5E shows the carp pvalb4_(Chr.A3) structure with the same conservation-based residue coloring as in Figure 5A, but now with the sidechains of what we designated as the conserved interior core residues highlighted in spheres format. This shows that there are basically two halves with conserved core residues, with the half that includes the Ca^2+^ binding domains showing major similarities with other EF-hand domain molecules (indicated by the black coloring), and the other half being more specific to parvalbumins (indicated by the gray coloring). Figure 5F is similar to Figure 5E, except that all the core residues are shown in sticks format and element coloring; this shows that they are all hydrophobic, except for the R75-E81 residue pair that is connected by polar sidechain interactions (indicated by dashed yellow lines). It was shown that if the R75-E81 salt bridge was deleted by mutations in rat α-parvalbumin, the overall molecular structure did not change but the flexibility and solvent accessibility of the AB loop increased [97]. In our estimation, the high conservation of this mostly buried R75-E81 suggests that the parvalbumin function involves some kind of plasticity/mechanism that may involve the inner core.

#### 3.5.4. The Conserved Core Residues Have Similar Positions in Those Structures of α-Parvalbumins, Oncomodulins, and Non-α/Non-Oncomodulin Parvalbumins That Are Determined by X-ray Crystallography

The core residue positions and orientations are very well conserved between structures determined by X-ray crystallography, whether being α-parvalbumin, oncomodulin, or non-α/non-oncomodulin parvalbumin. For visual clarity, in Figure 6, we separately depict what in Figure 5E,F are the “upper half” and “lower half” of conserved core residues. In Figure 6A it is shown that the upper half core residues superimpose very well between structures determined by X-ray crystallography, as does the (F/Y)57 residue located at the molecule surface (for discussion of this residue see below). For unknown reasons, the same residues have more variable positions and orientations in structures determined by solution NMR (Figure 6B). Similarly, the antiparallel β-strand-type polar interactions between the main chain amide- and carbonyl-groups of I58 and I97 are well conserved between all investigated X-ray crystallography structures and most investigated solution NMR structures (Figure 6C shows a single example), but in the solution NMR structures for human oncomodulin (1TTX) and Atlantic cod pvalb2 (2MBX) the I58 and I97 orientations are slightly different and only a single polar connection is recognized by the software (Figure 6D). For the bottom half of the conserved core residues, a similar phenomenon is observed, with only those determined by X-ray crystallography superimposing well (Figure 7A), while those in NMR-determined structures even show differences in the network of core phenylalanines and in the R75-E81 bond (examples in Figure 7B).

We have not been able yet to distinguish a clear pattern in how the NMR structures are different from the X-ray crystallography structures, as the NMR structures (even if including bound Ca^2+^ ions, as is the case in the structures depicted in our figures) are also different from each other. Because of their reproducibility, we assume that at least the X-ray crystallography structures are representative of a biological state. The various NMR structures may comprise a combination of different biological structural states related to function and technical artifacts. We refrain here from a further discussion of this topic, except for stressing that carefulness should be applied when basing functional plasticity models on differences between NMR and X-ray crystallography observations (e.g., [98]).

#### 3.5.5. The Conserved Residues at the Molecular Surface

Figure 8 shows surface representations of carp pvalb4_(Chr.A3) from two different angles, with gray and black representing conserved residues as in Figure 3 and Figure 5A. Most of the residues are hydrophilic, and among the hydrophobic residues, F57 has the biggest sidechain surface exposure; however, in some parvalbumins, Y57 is found instead (Figure 3 and Appendix A), so the aromatic character seems to be the main feature which is being selected at position 57. Most of the conserved surface residues are located at or near the Ca^2+^ binding sites (the molecule orientation in Figure 8A is as in Figure 5A). For some other residues, such as K27, it is not immediately clear why they are selected, and it might be for interacting with other (yet unknown) molecules.

### 3.6. Comparing the α-Parvalbumins, Oncomodulins, and Other Parvalbumins

In Figure 3, the residues that are characteristic of α-parvalbumins, oncomodulins, and non-α/non-oncomodulin parvalbumins are highlighted with magenta, green, and blue circles, respectively. In Figure 9A, their locations in the parvalbumin molecule are shown, revealing that most but not all reside in the outer region of the molecule.

#### 3.6.1. Characteristic Features of α-Parvalbumins

We distinguished the following 12 residues as characteristic of α-parvalbumins: M2, K12, K13, F18, (K/R)36, (V/I)66 (not F), K68, G69, G74, D76, K91, and E108. Five of these residues are basic, explaining why α-parvalbumins are less acidic than the other parvalbumins. Intriguing is the absence of F66, which in the other parvalbumins forms part of the inner core of phenylalanines (Figure 9B). The surface location of most of the 12 α-lineage characteristic residues (Figure 9C–E), and the fact that they have been well conserved for hundreds of millions of years, suggest that they may be involved in binding other molecules. Of the 12 residues, only K68 in α-helix D is conserved in all the 45 α-parvalbumins that we compared, and is Q68 in almost all of the other investigated parvalbumins (Appendix A). The K68 sidechain forms a salt bridge with the sidechain of α-parvalbumin-specific D76 in the DE-loop at the surface of the molecule (Figure 9E). Another pronounced difference from other parvalbumins is the possession of F18 instead of C18, although a very few α-parvalbumins do have C18 and a very few non-α parvalbumins lost C18 (Appendix A). The C18 residue can be used for cysteine-bridged homodimer formation [99,100], but that does not seem to be the natural situation [10,95,101], and, instead, C18 may have a redox sensory function [102]. Figure 9D shows that both C18 and F18 insert into the space between the α-helices A and B, and especially F18 may stabilize the contact between these helices.

As for the expression in ray-finned fishes, it is of note that the transcription of α-parvalbumin in gar and pvalb7 α-parvalbumin in pike is high in muscle (Appendix A). This agrees with pvalb7 in pike being reported as the abundant parvalbumin protein in muscle [19]. We are not aware that fish α-parvalbumins can induce allergy reactions. In some other teleost fish species, there are hardly *pvalb6* or *pvalb7* transcripts in muscle, and it is hard to distinguish a consistent pattern; however, *pvalb6* or *pvalb7* tend to be higher expressed in the brain than transcripts of the other parvalbumins (Appendix A).

#### 3.6.2. Characteristic Features of Oncomodulins

We deemed the following three positions as characteristic of oncomodulins: (D/E)17, (D/E/N)52, and (R/K)69. However, none of these three features is completely conserved (Appendix A). If the concept that the C18 residue is a redox sensor is true [102], the neighboring (D/E)17 residue (Figure 9D) may have an effect on that. At least in some, and maybe in all oncomodulins, (D/E/N)52 at the start of the CD loop and (R/K)69 near the C-terminal end of α-helix D make a polar contact (Figure 9E), and this may have an effect on the flexibility of the CD-loop for binding Ca^2+^. It should be noted that at position 69 also, α-parvalbumins and non-α/non-oncomodulin parvalbumins have their own rather well-conserved residues (Figure 3 and Appendix A). Of the different phylogenetic tree methods applied (Appendix A), only the maximum likelihood method convincingly recognized elephant shark pvalb-6258 as an oncomodulin, but its oncomodulin identity was confirmed by genomic location (Figure 2) and conservation of the signature motif (Appendix A). As for expression in teleost fish, we have not been able to distinguish a consistent tendency for expression in a certain tissue for either oncomodulin *pvalb8* or *pvalb9* (Appendix A).

**Figure 9 biology-11-01713-f009:**
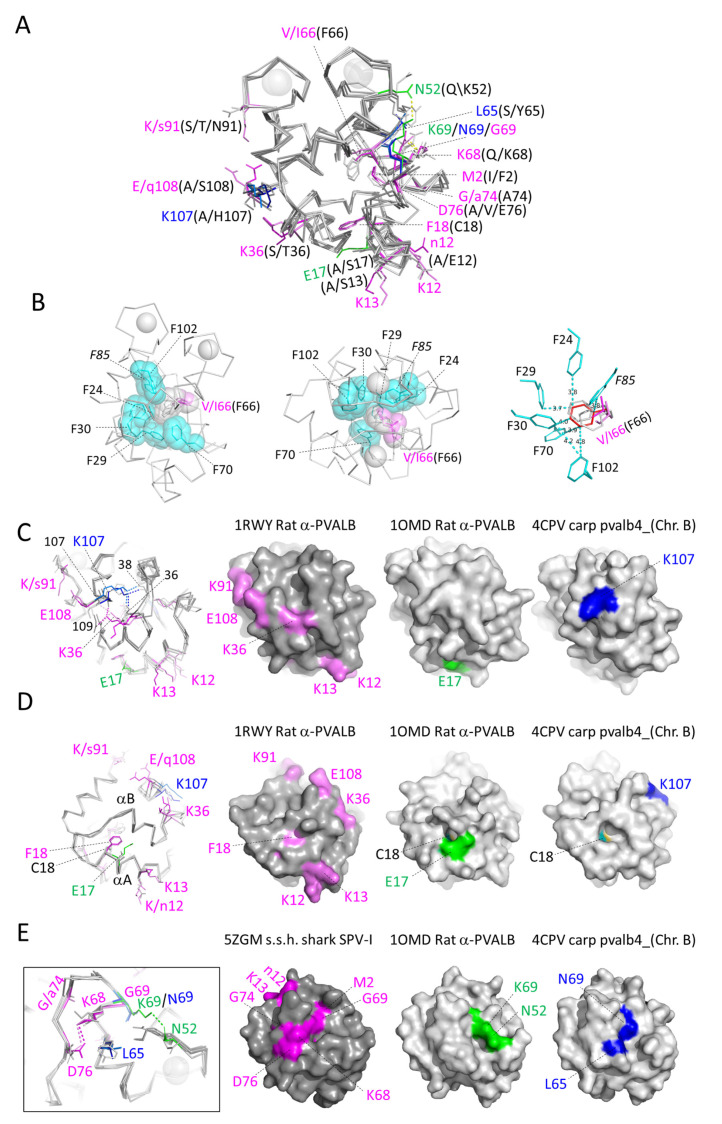
Structural locations of residues that are rather characteristic for the α-parvalbumins, oncomodulins, or other parvalbumins. Sidechains of residues at positions with category-specific features are shown in sticks format, in color when they are category-specific and otherwise in gray. Superimposition of X-ray crystallography structures of α-parvalbumin (lineage-specific sidechains in violet, rat α-parvalbumin [1RWY]; magenta, spotless smooth-hound shark SPV-I [5ZGM]), oncomodulin (sidechains in green, rat oncomodulin [1OMD]), and other parvalbumins (sidechains in marine blue, chicken ATH [3FS7]; blue, common carp pvalb4_[Chr.A3] [4CPV]). (**A**) In this figure, all the residues at the relevant positions are named. Most of the category-specific residues are located outside of the molecules, but the (V/I/F)66 and (C/F)18 residues are among the exceptions and reside (mostly) within the inner part of the molecules. (**B**) In α-parvalbumins one phenylalanine, at position 66, is lacking from an inner core set of phenylalanines compared to the other parvalbumin categories. Apart from the residues at position 66, this figure only shows common carp pvalb4_(Chr.A3) residues. F85 is only rather well conserved throughout parvalbumins (it is not highlighted with gray or black in Figure 3), which is why it is indicated in Italic font. Apart from in sticks format, in the two figures at the left (which show the same from different angles) the relevant sidechains are also highlighted in transparent spheres format. In the figure at the right, with carp pvalb4_(Chr.A3) F66 in red, the distances in Å between several atoms of the carp pvalb4_(Chr.A3) phenylalanines are indicated. (**C**) The sidechain of α-parvalbumin-specific K36, located in the BC-loop, makes polar contacts (dashed lines) with the main chains of residues 107 and 109 at the C-terminus of the molecule. The sidechain of non-α/non-oncomodulin category-specific K107, in α-helix F, makes polar contacts with the main chains of residues 38 and 36 in the BC-loop. On the right, with a similar molecule orientation and coloring as in the superimposition figure on the left, the surface structures of one molecule per category are shown. (**D**) At position 18, α-parvalbumins have a phenylalanine whereas most other parvalbumins have a cysteine. In either case, the residue points from near the C-terminal end of α-helix A to the molecule interior under α-helix B and is mostly covered. In the surface figures at the right, the C18 residue is colored with cyan for carbon and yellow for sulfur. (**E**) In α-parvalbumins, polar contacts are formed between the sidechains of lineage-specific K68 in α-helix D and D76 in the DE-loop. In rat oncomodulin, polar contacts are formed between the sidechains of lineage-specific residues N52 in the CD-loop and K69 in α-helix D.

#### 3.6.3. Characteristic Features of Non-α/Non-Oncomodulin Parvalbumins

As characteristics of the non-α/non-oncomodulin parvalbumins, we deemed the three residues L65, N69, and K107, although none of them is perfectly conserved (Appendix A). The K107 residue at the end of helix F makes polar contacts with the main chains of residues 36 and 38 in the BC-loop and possibly serves to strengthen this connection and/or to build a characteristic surface patch (Figure 9C). The L65 and N69 residues are connected as they are roughly one helix turn apart in α-helix D (Figure 9E). The L65 sidechain is exposed at the molecule surface, providing a hydrophobic patch with unknown function (Figure 9E).

### 3.7. Characteristic Features of the Individual Non-α/Non-Oncomodulin Parvalbumins in Teleost Fish

The teleost molecules pvalb5 and pvalb10 can each be identified by phylogenetic tree analysis as well as by having a set of characteristic residues. The molecules pvalb1-to-4, on the other hand, despite being located on two different chromosomes for more than 300 million years, do not have clearly distinguishing features, and even the few residues that are somewhat specific for one of these four parvalbumins can also be found in some individual members of the other three parvalbumins. Because of their location (Figure 2), it is tempting to speculate that teleost *pvalb5* and *pvalb10* are duplicates of the same gene resulting from the teleost-specific WGD; however, there is no evidence for that, neither by phylogenetic tree analysis (Appendix A) nor by sharing of characteristic residues (Appendix A). The structural positions of the below-discussed characteristic residues are highlighted in Figure 10. Although both pvalb5 and pvalb10 lineages could be recognized in the phylogenetic tree analysis by Mukherjee et al., 2021 [26], to our knowledge the current study is the first to analyze their characteristics.

#### 3.7.1. Characteristic Features of Teleost Fish Parvalbumins pvalb1-to-4

The lack of clearly distinguishing features (Appendix A), the shared expression in muscle tissue (Appendix A), and the variable gain and loss of gene copies while always keeping some copies (Figure 2), suggest that the pvalb1-to-4 molecules have nearly identical functions. The sequence data seem to be consistent with a model in which the multiple copies provide teleost fish with a possibility to fine-tune, possibly between different muscle tissues, a conserved function that may hardly change. There are only very subtle differences between the four parvalbumins as highlighted in Figure 3 and Appendix A. Pvalb1, in contrast to pvalb2, -3, and -4, tends not to have a lysine but an asparagine at position 54 in the CD loop (for the location see Figure 5C and Figure 10), and although the sidechain at this position points away from the metal ion it might indirectly affect its binding. Pvalb2 commonly has A12 whereas pvalb3, which is encoded from the same chromosome (Figure 2), has no alanine and tends to have a threonine (Appendix A) at this position in α-helix A which borders the molecule exterior; maybe it causes this molecular region to interact differently with other molecules, although A12 is not unique to pvalb2 (as exemplified by carp pvalb4_[Chr.A3] in Figure 10). Pvalb4 has the characteristic residues (D/E)16 and K19, which are adjacent (roughly one turn apart) residues in α-helix A at the molecule exterior (Figure 10); at least as a pair they distinguish pvalb4 from pvalb1, -2, and -3 (Appendix A) and they may impact the interaction with other molecules.

Among the teleost parvalbumins, so far, only the teleost pvalb1-to-4 molecules have been shown to cause allergic reactions (for some references see Appendix A) and to them, the bulk of teleost parvalbumin research has been dedicated. There have been several studies that, together, mapped a variety of epitopes targeted by allergic IgE antibody responses (e.g., [87,103]), but the unusually high protein abundance [13,104] and stability [105,106] of these fish parvalbumins may be more responsible for their high allergenicity than the (potentially special) character of the allergenic epitopes which are quite diverse between reports. The abundance of pvalb1-to-4 in teleost muscle speaks to their functional importance and their stability probably derives in a large part from the hydrophobic core with multiple phenylalanines (see Figure 9B). It remains to be determined why teleost fish have multiple molecules of the pvalb1-to-4 family.

#### 3.7.2. Characteristic Features of Teleost Fish pvalb5

Rather characteristic for teleost pvalb5 are P21, non-K at position 38, a non-hydrophobic residue at position 43, E76, C103, and (I/L/M)106 (Figure 3 and Appendix A). Among these, the most intriguing probably is the lack of a hydrophobic residue at position 43 in helix C in the molecule interior which sets them apart from all other investigated parvalbumins (Appendix A). Without obvious pvalb5 in species other than teleost fish, it seems that this lineage was established rapidly in early teleost fish, after which the unique features have been rather well conserved. The unique residues suggest that it has a unique function, but *pvalb5* expression levels are very low and difficult to attribute to a specific tissue across teleost species (Appendix A). Furthermore, some fish lost the *pvalb5* gene (Figure 2), also arguing against its importance. Future analysis will have to determine pvalb5 function.

#### 3.7.3. Characteristic Features of Teleost Fish pvalb10

Other than in teleost fish, *pvalb10* genes can also be found in primitive ray-finned fish like bichir and gar (Figure 2 and Figure 4, Appendix A). The pvalb10 molecules, especially if only looking at those in teleost fish, have a rather large set of characteristic residues that set them apart from other parvalbumins in teleost fish: D4, S9, (K/R)37, E44, K72, W99, and (S/T)109 (Figure 3 and Appendix A). The most notable one, found in most but not all teleost pvalb10, is W99 in the N-terminal part of α-helix F, which based on the other parvalbumin structures would be expected to have its large hydrophobic sidechain exposed to the molecule exterior and to modify the connection between the two Ca^2+^ binding loops. Some teleost fish lost the *pvalb10* gene (Figure 2) and in some *pvalb10* expression is very low; however, in those of the investigated fish that have a considerable level of *pvalb10* expression, the brain is consistently among the tissues with the highest expression (Appendix A). As for pvalb5, future studies will have to determine pvalb10 function.

## 4. Conclusions

The present study comprises a comprehensive analysis of parvalbumin sequences. Actually, we originally set out to address another question about fish parvalbumins but found that at first, the here presented type of study was necessary before any other question on differences in parvalbumin function or allergenicity between fish species could be addressed. We hope and expect that this will also be useful for other scientists. By looking at a variety of parameters, in teleost fish, we could distinguish between pvalb1-to-10 genes of which, for several of these genes, the copy number can vary. Especially for teleost pvalb1-to-4, which are highly expressed in muscle and include well-known allergens, gene identification is complicated and their summarization as “microheterogeneity” [86] or “high isoform complexity” [87] is understandable and in many studies would be sufficient from a practical point of view. However, for some scientific questions, better gene identification and understanding of the molecules at the sequence level is necessary, for which we here provide a frame of reference.

Apart from teleost fish parvalbumins, the present study also helps to understand the evolution of parvalbumins throughout jawed vertebrates by identifying commonly conserved residues as well as residues characteristic of α-parvalbumins, oncomodulins, and other parvalbumins. To the best of our knowledge, this is also the first report of non-α/non-oncomodulin parvalbumins in some mammals. Future studies will have to clarify the functions of the different teleost parvalbumins, as well as clarify the structure-function relationships that distinguish the different lineages of parvalbumin.

## Figures and Tables

**Figure 1 biology-11-01713-f001:**
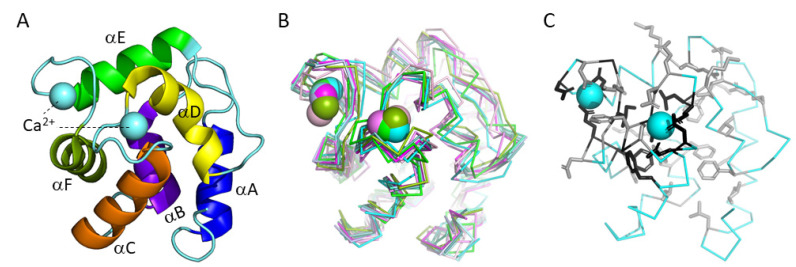
Parvalbumins have six α-helices A-to-F and two “EF-hand” domains for binding Ca^2+^ ions (indicated as spheres). (**A**) The structure, in cartoon format, of common carp pvalb4_(Chr.A3) (PDB accession 4CPV) [10], which was the first parvalbumin of which the structure was elucidated [11]. Different α-helices are in different colors. (**B**) Superimposition of various parvalbumin structures, in ribbon format, reveals a common structure. Light pink, human α-parvalbumin (1RK9); pink, pike pvalb7 (2PAS); magenta, spotless smooth-hound shark SPV-I α-parvalbumin (5ZGM); green, human oncomodulin (1TTX); splitpea green, chicken CPV3-oncomodulin (2KYF); soft purple, chicken ATH (3FS7); cyan, Atlantic cod pvalb2 (2MBX); green cyan, pike pvalb3 (1PVB); aquamarine, common carp pvalb4_(Chr.A3) (4CPV); light teal, spotless smooth-hound shark SPV-II (5ZH6). (**C**) Most of the conserved EF-hand family residues belong to the Ca^2+^ binding regions (see also Figure 5). The structure, in ribbon format, of common carp pvalb4_(Chr.A3) (PDB accession 4CPV), shows in black those residues that are well conserved throughout EF-hand domain family molecules and in gray other residues that are well conserved throughout parvalbumins; the sidechains of these residues are shown in sticks format.

**Figure 2 biology-11-01713-f002:**
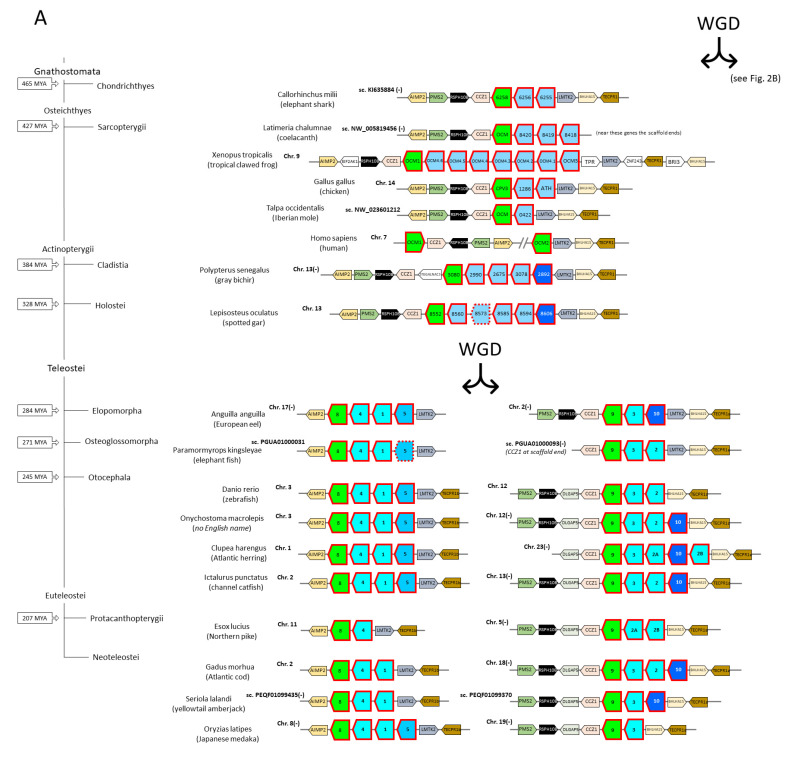
Parvalbumin genes situate in ohnologous regions derived from whole genome duplications (WGD), in humans located on Chr. 7 ((**A**); first page) and Chr. 22 ((**B**); second page)), and are rather well but not perfectly conserved throughout jawed vertebrates. The parvalbumin genes are indicated with red-lined boxes, which are colored magenta for α-parvalbumins, green for oncomodulins, and different kinds of blue for pvalb10, teleost pvalb5, and teleost pvalb1-to-4. Neighboring non-parvalbumin genes are indicated by lower boxes with their name abbreviations inside. For the full names of the parvalbumin genes and their encoded sequences see Appendix A. Box with dashed line, probable pseudogene; sc., scaffold. At the left, the evolution of relevant species clades is shown folowing [77] and [79]; MYA, million years ago. The genomic information was derived from either the following Ensembl datasets: Elephant shark, Callorhinchus_milii-6.1.3; chicken, GRCg6a; human, GRCh38.p13; spotted gar, LepOcu1; elephant fish, PKINGS_0.1; zebrafish, GRCz11; Atlantic herring, Ch_v2.0.2; channel catfish, lpCoco_1.2; Northern pike, Eluc_v4; Atlantic cod, gadMor3.0; yellowtail amberjack, Sedor1; Japanese medaka (HdrR strain), ASM223467v1; or from the following NCBI datasets: Coelacanth, LatCha1; Tropical clawed frog, UCB_Xtro_10.0; Iberian mole, MPIMG_talOcc4; gray bichir, ASM1683550v1; European eel, fAngAng1.pri; *Onychostoma macrolepis*, ASM1243209v1. Many of the indicated names for parvalbumin genes of elephant shark, chicken, gray bichir, and spotted gar are the last four numbers of gene names used in the respective databases. The symbol (-) indicates that the genomic region is depicted in opposite direction from the respective database report.

**Figure 5 biology-11-01713-f005:**
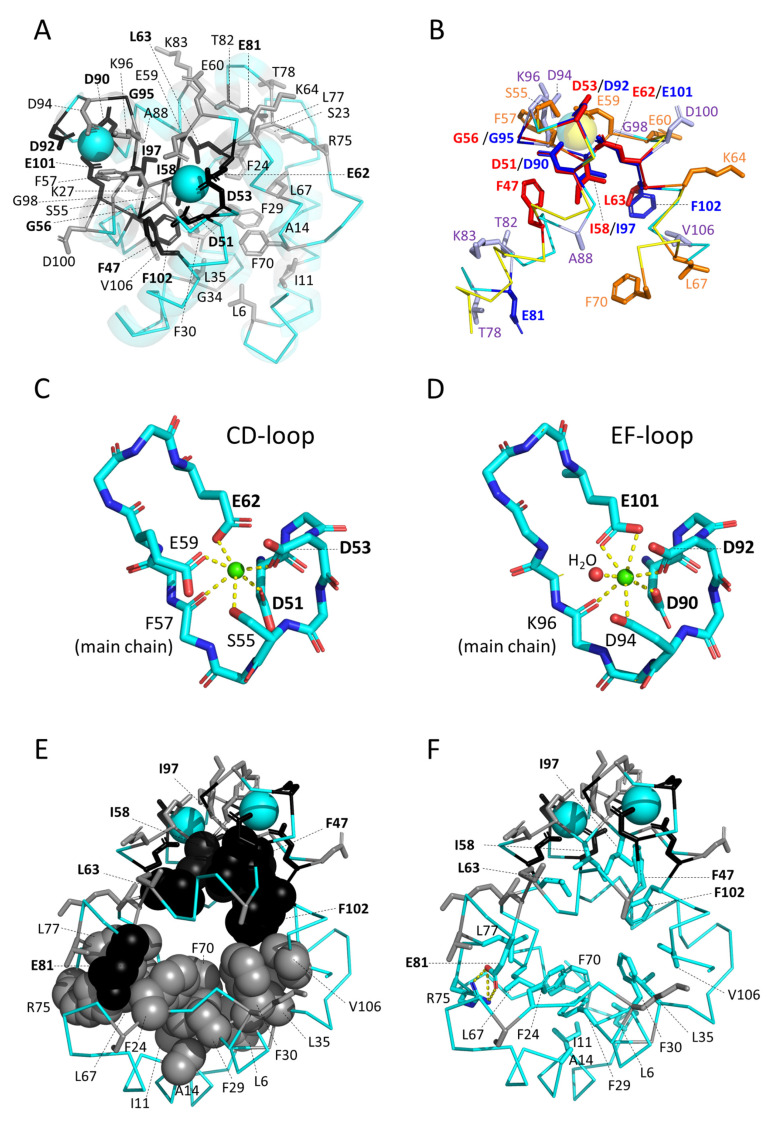
Structural positions of residues that are conserved in EF-hand domain family molecules and parvalbumins. (**A**) The structure of common carp pvalb4_(Chr.A3) (PDB accession 4CPV) with highlighting in sticks format the residues conserved in EF-hand domain family molecules (black) or more specifically in parvalbumins (gray). This figure is similar to Figure 1C except for the naming of the residues and an additional indication of secondary structures in transparent cartoon format. (**B**) Superimposition of the CD (default yellow) and EF (default cyan) regions of the same molecule as in (**A**), while highlighting their conserved residues in sticks format (red and orange for CD region residues, and blue and light-purple for EF region residues, respectively, that are conserved in EF-hand domain family molecules or more specifically parvalbumins, respectively) shows that the two regions are related and bind Ca^2+^ in a similar but also different manner. Matching pairs are D51/D90, D53/D92, G56/G95, I58/I97, and E62/E101. As for acidic residues that directly bind Ca^2+^, notable differences are the D94 residue in the EF-region (which matches conserved CD-region residue S55) and the E59 residue in the CD-region (which matches conserved EF-region residue G98). (**C**,**D**) The polar interactions with Ca^2+^ in the CD-region and EF-region, respectively, of carp pvalb4_(Chr.A3) (PDB accession 4CPV). All the binding residues are conserved (gray or black in (**A**)). In the EF-region, Ca^2+^ also binds a water molecule, which is usual among parvalbumins. (**E**) Conserved residues of the inner core (black and gray coloring as in (**A**)) seem to be divided into roughly two regions, one associated with the Ca^2+^ binding sites which consists of residues shared with other EF-hand domain molecules and another at the molecule opposite which consists mostly of residues that are more specific for parvalbumins. The (somewhat arbitrarily defined) core residues have their sidechains highlighted in spheres format and are the only ones that are named. The structure is the same as in (**A**) but shown from a different orientation. (**F**) The same figure as in (**E**) but with the core residues shown in sticks format and element coloring (aquamarine for carbon), revealing that most are hydrophobic. Exceptions are the R75 and E81 residues that form a salt bridge (dashed yellow lines indicate polar contacts).

**Figure 6 biology-11-01713-f006:**
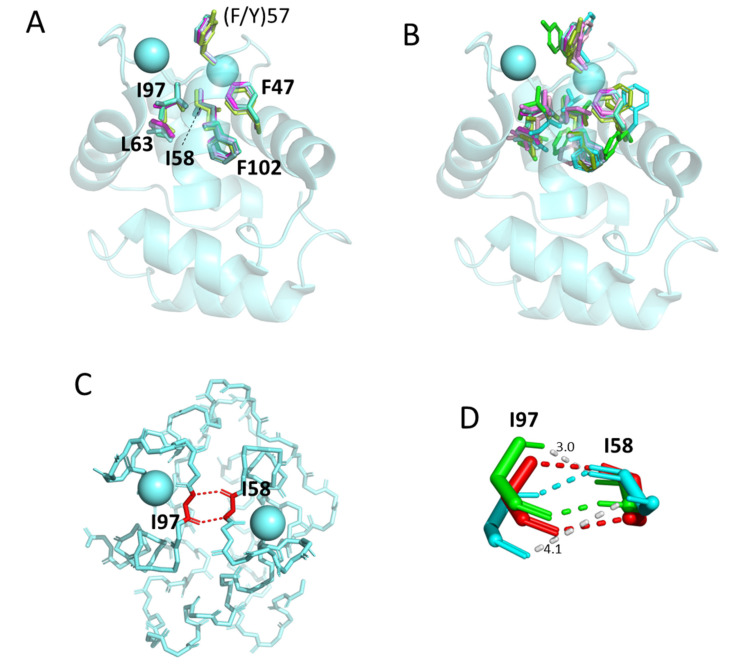
The conserved core residues in the proximity of the Ca^2+^ binding sites show a well-conserved orientation among parvalbumin structures determined by X-ray crystallography but less so among structures determined by NMR. (**A**) Superimposition of X-ray crystallography structures of α-parvalbumin (magenta, spotless smooth-hound shark SPV-I [5ZGM]), oncomodulin (limon, rat oncomodulin [1OMD]), and other parvalbumins (soft purple, chicken ATH [3FS7]; green cyan, pike pvalb3 [1PVB]; aquamarine, common carp pvalb4_[Chr.A3] [4CPV]; light teal, spotless smooth-hound shark SPV-II [5ZH6]). (**B**) As (**A**), but with additional superimposition of structures determined by solution NMR of α-parvalbumin (light pink, human α-parvalbumin [1RK9]; pink, pike pvalb7 [2PAS]), oncomodulin (green, human oncomodulin [1TTX]; splitpea green, chicken oncomodulin [CPV3] [2KYF]), and other parvalbumin (cyan, Atlantic cod pvalb2 [2MBX]). (**C**) The beta-strand type connection between conserved residues I58 and I97 is well-conserved between all investigated structures determined by X-ray crystallography and most structures determined by solution NMR. The structure shown here, only showing the main chains in sticks format, is of common carp pvalb4_(Chr.A3) (4CPV). (**D**) In the solution NMR structures for human oncomodulin (1TTX) (green) and Atlantic cod pvalb2 (2MBX) (cyan), the I58 and I97 orientations are somewhat distorted compared to other investigated structures (here exemplified by common carp pvalb4_[Chr.A3] [4CPV] in red; polar contacts are indicated with dashed lines in a color matching the relevant structure depiction) and software only recognizes one and not two polar bonds between the main chains. The gray dashed lines, measured in Å, indicate distances between relevant 1TTX and 2MBX atoms where in other parvalbumin structures polar contacts are recognized.

**Figure 7 biology-11-01713-f007:**
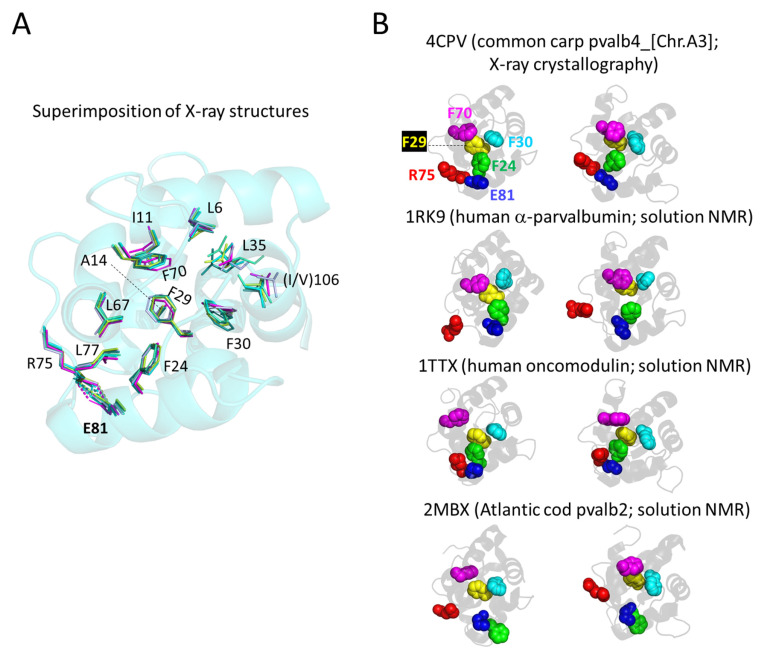
The conserved core residues at the side of the molecules opposite of the Ca^2+^ binding sites (the “bottom” conserved residues in Figure 5E,F) also display rather well-conserved orientations among parvalbumin structures determined by X-ray crystallography but less so among structures determined by NMR. (**A**) Superimposition of X-ray crystallography structures of α-parvalbumin (magenta, spotless smooth-hound shark SPV-I [5ZGM]), oncomodulin (limon, rat oncomodulin [1OMD]), and other parvalbumins (soft purple, chicken ATH [3FS7]; green cyan, pike pvalb3 [1PVB]; aquamarine, common carp pvalb4_[Chr.A3] [4CPV]; light teal, spotless smooth-hound shark SPV-II [5ZH6]). The view is from “above” (from the direction of the Ca^2+^ ions) and shows the transparent cartoon format structure of carp pvalb4_(Chr.A3); for the other molecules, only the sidechains of the highlighted residues are shown in sticks format. (**B**) In the structures determined by NMR, the otherwise conserved core formed by the four phenylalanines F24, F29, F30, and F70 can be absent, as well as the salt bridge between R75 and E81. The figures at the left and right show the same structure from somewhat different angles. At the top, for comparison, the common carp pvalb4_(Chr.A3) structure is shown as determined by X-ray crystallography, and the other figures depict human α-parvalbumin (1RK9), human oncomodulin (1TTX), and Atlantic cod pvalb2 (2MBX) in the same orientation (by superimposition function) as the above-shown carp pvalb4_(Chr.A3) structure.

**Figure 8 biology-11-01713-f008:**
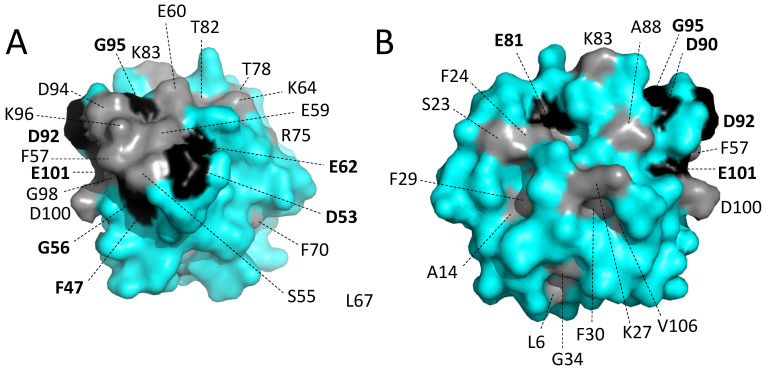
Surface location of residues conserved throughout parvalbumins. The figures (**A**,**B**) are surface format depictions of common carp pvalb4_(Chr.A3) (4CPV) from different angles. Black and gray coloring refer to conservation among EF-hand domain molecules or more specifically in parvalbumins, respectively (as in Figure 3 and Figure 5A).

**Figure 10 biology-11-01713-f010:**
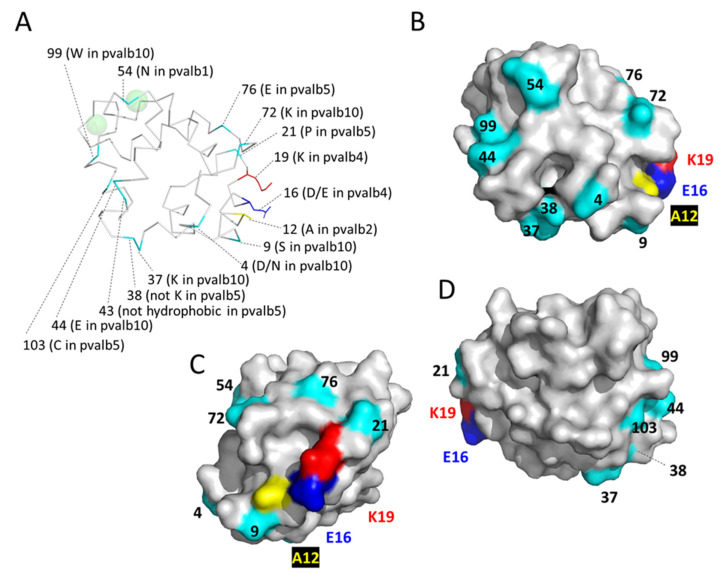
Structural locations of residues characteristic of individual non-α/non-oncomodulin teleost parvalbumins. The structure of common carp pvalb4_(Chr.A3) (4CPV) is shown is ribbon format (**A**) and in surface format from different angles (**B**–**D**), with the colors yellow, blue, and red for characteristic residues that are present in the depicted pvalb4 molecule and cyan for the other highlighted residues. In (**A**), sidechains are only shown for the pvalb4 characteristic residues E16 and K19 and the pvalb2 characteristic residue A12 (which is also present in the depicted pvalb4 molecule). The listed characteristics are for the teleost molecules (residue 9 in pvalb10 of the non-teleost fishes gar and bichir is not a serine; see Appendix A).

## Data Availability

Sequences with their database accessions are provided in Appendix A.

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
