# Peer review of "Comprehensive Sequence Analysis of Parvalbumins in Fish and Their Comparison with Parvalbumins in Tetrapod Species"

_biology, 2022, doi:10.3390/biology11121713_

Round 1
Reviewer 1 Report
This study by Dijkstra and Kondo describes a new phylogenetic and structural survey of the parvalbumin gene family. The study follows closely on the heels of a recent one by Mukherjee et al 2021, but I appreciated the care with which the nomenclature and nuances of the gene family names were discussed here by the authors. The paper is organized and clear and will provide an updated look at this gene family for those interested.
Two points that I would consider addressing:
1) Line 163: Alignments were done by hand? I realize the gene family is quite conserved, but would recommend either providing the alignment as supplement (not just the sequence but the alignment) or using a standard algorithm so that the analyses can be replicated. At present, there is no way to verify analyses without the author's hand alignment.
2) Some of the statements of the alignment analysis might benefit from a slightly expanded pool of outgroup sequences. Including a more diverse array of mammals and birds, as well as including some squamate or crocodilian sequences (there appear to be some in GenBank) might give more specificity to some of the teleost profiling. Completely at author's discretion, just a suggestion.
Author Response
We thank the Reviewer for his/her careful analysis and the appreciation of our work.
As for comment 1 ("Line 163: Alignments were done by hand? I realize the gene family is quite conserved, but would recommend either providing the alignment as supplement (not just the sequence but the alignment) or using a standard algorithm so that the analyses can be replicated. At present, there is no way to verify analyses without the author's hand alignment."):
Our alignment figure was already provided as Supplementary file 3. To make this easier to realize we have now added a sentence to the materials and methods paragraph 2.3. Why we prefer to align by hand is explained in the there referenced citation No. 75. It has to do with allowing intron-exon organization, relatedness, and structure arguments into the decision making.
Having said that, among the parvalbumin sequences there are hardly any gaps in the alignments (see Supplementary file 3) so that their alignment would be very similar regardless of the alignment method and the important conclusions would be the same.
Additionally, in Supplementary file 1 all the sequences are provided in a way that allows an expert to make them ready within 5 min for any type of computerized analysis.
As for comment 2 ("Some of the statements of the alignment analysis might benefit from a slightly expanded pool of outgroup sequences. Including a more diverse array of mammals and birds, as well as including some squamate or crocodilian sequences (there appear to be some in GenBank) might give more specificity to some of the teleost profiling. Completely at author's discretion, just a suggestion.")
The Reviewer is right that a more thorough analysis of parvalbumins in tetrapods (or Chondrichthyes) would be interesting. However, such analysis and discussion would also make the paper much larger. In the current paper, we mainly focus on ray-finned fish because their parvalbumins cause allergy problems. The parvalbumin sequences in the other animal clades are only introduced as outgroups and for getting a grasp on deep parvalbumin evolution. For that purpose, in our opinion, we have included enough parvalbumin sequences from non-fish species already.
We hope that the Reviewer can accept our point of view and thank him/her again for the time and positive attitude.
Sincerely,
Also on behalf of Yasuto Kondo,
Hans Dijkstra
Reviewer 2 Report
Dear authors
Your manuscript about sequence analysis of parvalbumins showed an extensive analysis of parvalbumin structures and evolution of several species. Important species have been included in the analysis and conclusions are based on the obtained results.
The discussion section is quite extense, however all analysis were mentioned properly and authors tried to keep a logic flow.
Line 11: multiple different? in which way do authors mean these differences? sequence, lineage, structural? Not all fish have alpha and beta parvalbumins.
Author Response
We thank the Reviewer for his/her time and the very positive comments.
As for the comment ("Line 11: multiple different? in which way do authors mean these differences? sequence, lineage, structural? Not all fish have alpha and beta parvalbumins.")
We have now written this clearer into "Fish have multiple parvalbumins with quite similar sequences, making a comparison between fish species complex."
Because this fragment is in the "Simple Summary," we probably should not start discussing alpha and beta lineages.
Sincerely,
Also on behalf of Yasuto Kondo,
Hans Dijkstra